# Production of Lactic Acid from Carob, Banana and Sugarcane Lignocellulose Biomass

**DOI:** 10.3390/molecules25132956

**Published:** 2020-06-27

**Authors:** Hassan Azaizeh, Hiba N. Abu Tayeh, Roland Schneider, Augchararat Klongklaew, Joachim Venus

**Affiliations:** 1Institute of Applied Research, The Galilee Society, P.O. Box 437, Shefa-Amr 20200, Israel; hebaabutayeh@yahoo.com; 2Department of Environmental Science, Tel Hai College, Upper Galilee 12208, Israel; 3Faculty of Natural Sciences, University of Haifa, Abba Khoushy Ave 199, Haifa 3498838, Israel; 4Leibniz Institute for Agricultural Engineering and Bioeconomy, Max-Eyth-Allee 100, 14469 Potsdam, Germany; rschneider@atb-potsdam.de (R.S.); jvenus@atb-potsdam.de (J.V.); 5Interdisciplinary Program in Biotechnology, The Graduate School, Chiang Mai University, Chiang Mai 50200, Thailand; augchararat.taey@gmail.com

**Keywords:** lactic acid, lignocellulose, carob biomass, banana peduncle biomass, sugarcane biomass, fermentation

## Abstract

Lignocellulosic biomass from agricultural residues is a promising feedstock for lactic acid (LA) production. The aim of the current study was to investigate the production of LA from different lignocellulosic biomass. The LA production from banana peduncles using strain *Bacillus coagulans* with yeast extract resulted in 26.6 g LA·L^−1^, and yield of 0.90 g LA·g^−1^ sugars. The sugarcane fermentation with yeast extract resulted in 46.5 g LA·L^−1^, and yield of 0.88 g LA·g^−1^ sugars. Carob showed that addition of yeast extract resulted in higher productivity of 3.2 g LA·L^−1^·h^−1^ compared to without yeast extract where1.95 g LA·L^−1^·h^−1^ was obtained. Interestingly, similar LA production was obtained by the end where 54.8 and 51.4 g·L^−1^ were obtained with and without yeast extract, respectively. A pilot scale of 35 L using carob biomass fermentation without yeast extract resulted in yield of 0.84 g LA·g^−1^ sugars, and productivity of 2.30 g LA·L^−1^·h^−1^ which indicate a very promising process for future industrial production of LA.

## 1. Introduction

Lignocellulosic biomass from agricultural crop residues is a promising raw material for lactic acid (LA) production due to its abundant availability, and its utilization can decrease environmental contamination and pollution. Lignocellulosic biomass is considered a cheap, abundant, and renewable raw material for the production of different by-products such as biofuels, biomolecules, biomaterials, and bioenergy. Therefore, it represents a more sustainable alternative resource. Lignocellulosic residues are composed of cellulose, hemicellulose, and lignin. Pretreatments are required, followed by enzymatic hydrolysis processes, where the released sugars are utilized for the production of economical products including bioethanol, bacteriocins, lipoteichoic acid, probiotics, biogas and LA [1,2,3,4]. The bioconversion of lignocellulose to LA is an important alternative for its valorization to produce LA to be utilized in food, cosmetic and pharmaceutical industry [4,5]. The global demand for LA is increasing, with its demand expected to reach 1,960 kilotons by 2025 [6,7,8]. Different companies around the world produce about 90% of LA through microbial fermentation of crops (mainly corn) and lignocellulosic biomass [5]. The numerous applications of LA have made it one of the most important products and its biotechnological production includes integration of pretreatment processes, enzymatic hydrolysis and fermentation. The scaled up processes are aimed at decreasing energy consumption and costs without affecting LA purity [9,10,11].

The global interest in LA production is due to its utilization in various products including PLA, a biodegradable plastic to replace unbiodegradable synthetic plastic. The focus nowadays is on the use of different types of treatments used for the transformation of lignocellulosic biomass into suitable substrates for industrial production. Therefore, several methods of fermentations and genetic modifications are underway in order to increase the production and the scale up of LA process [3,12,13]. The biodegradability and biocompatibility of PLA makes the polymer safe and ideal for its application in a wide range of industries, with increased environmental awareness worldwide promoting the development of such technologies for a sustainable approach [14]. 

Currently, most of the LA is manufactured in batch reactors; however, continuous fermentation could offer various advantages, especially in terms of productivity and operational costs. *Bacillus coagulans* strains have been effectively employed for the fermentation of various lignocellulosic biomass such as corn stover, agricultural straw, bagasse and other organic wastes for LA production [2,12,15,16]. Attention is increasingly focused on the utilization of cheap biomass especially bagasse to reduce environmental contamination and pollution.

Carob (*Ceratonia siliqua* L.) is a perennial leguminous, an evergreen plant widely grown in the Mediterranean region with great economic and environmental potential where it covers hills and large mountain areas of the arid and the semi-arid regions [17,18]. World production of carob is about 422,334 tons per year [19], where the pods consist of 90% pulp and 10% seeds by weight [20]. It is utilized as a raw material in different food industries including animal feed, cakes and yogurt production [21,22]. As well as in cosmetics and pharmacological industry and as drug delivery purposes [23]. The pods of carob are rich mainly with glucose 7–10%, fructose 10–12% and sucrose 34–42% [24,25]), used for the production of natural syrups where the residues of the lignocellulosic bagasse are mostly discarded as waste. Fruit syrup is used to treat various sicknesses and diseases including herpes, digestives system, as drug delivery matrix, in addition to bioethanol and biogas production [17,26,27,28]. (In addition, carob syrup is used for the preparation of cakes, food, molasses and the solid waste/bagasse is either discarded or used for animal feed. The bagasse waste is rich with sugars (44% of dry matter) and could represent an important and cheap lignocellulosic biomass to produce LA and bioethanol to be used as second generation energy [29]). To the best of our knowledge no research has been conducted to produce LA from carob bagasse.

Bananas are an important crop cultivated worldwide in the tropics and subtropics, where more than 106 million tons of fruits are produced every year to meet the increasing world demand [30,31]. The peduncles which represent ca 13% of the biomass are separated from the cluster and discarded as organic waste [32]. Some studies have used peduncles as a raw organic material for paper production [33], fiber boards, compost, agricultural fertilizers [34], an additive for burger preparations [35], and bioethanol production [32]. The generated lignocellulosic banana biomass is rich with various sugars mainly sucrose, fructose and glucose. The banana pseudo-stems are rich with cellulosic content of 42.2–63%, banana wastes were used for the release of fermentable sugars for LA production [36,37]. To the best of our knowledge no research was conducted in order to produce LA from banana peduncle bagasse. Another biomass of interest is sugarcane (*Sacchrum officinarum*), which is known with high amounts of lignocellulosic biomass remaining following sugar extraction as bagasse waste. In Brazil, for example, during the period 2015–2016, more than 666 million tons of sugarcane was produced, and for each ton approximately 250–270 kg of bagasse were generated as lignocellulosic wastes [38]. This abundant residue is an important bagasse waste for the production of second-generation bioethanol and LA to be utilized for PLA [38,39]. Both carob and banana are abundant in our region and produce high amounts of lignocellulose residues; therefore, it is important to test their potential to produce LA to be compared with sugarcane bagasse.

The main aim of the current study is to investigate the production of LA from different biomass residues including carob, banana peduncles and sugarcane bagasse. Subsequently, the effect of the hydrolysis on sugar contents, types and LA production were also evaluated in batch fermentation processes. For carob bagasse only, a 35 L pilot batch scale was performed as well.

## 2. Results and Discussion

### 2.1. Lignocellulose Biomass Analysis

The dry matter (DM) of the different lignocellulose biomass of banana, sugarcane and carob used for the LA production ranged between 85.8–93.2% (Table 1). The contents (*w*/*w*) of lignin in banana peduncle, and sugarcane biomass were relatively low, 6.16% and 2.79% respectively (Table 1). The contents (*w*/*w*) of the lignin of the carob biomass were high 28.4%, and the total sugars were 27.7% (Table 1). It is known that carob is rich in fiber, relatively low cellulose and hemicelluloses 18%, high levels of tannins 16–20%, antioxidants and phenolic compounds 2–20%, and relatively high levels of proteins 2.7–7.6% [23]. Our results have shown similar values where the cellulose and hemicelluloses of carob together were 19.35% of the total biomass which mainly consisted of cellulose (Table 1).

### 2.2. Fermentation of Banana Peduncles

Hydrolysis of the banana biomass at 10% and 15% DM using a mixture of Accellerase BG and Cellic CTec2 and fermentation process of a separate hydrolysis and co-fermentation (SHCF) with isolate A166 for LA production were carried out. It is well known that enzymatic hydrolysis presents advantages over chemical hydrolysis, since much less extreme conditions are used and negligible amounts of fermentation inhibitors such as furfural, 5-hydroxymethylfurfural, organic acids, and other lignin derivatives are usually generated [40,41]. Since the 10% DM of banana peduncles showed the same trend as the 15% DM, the results of 15% DM are the only presented data. Sugar types and content of banana peduncles biomass (15% DM) after enzymatic hydrolysis resulted in 32.6 g·L^−1^ total sugars (mono- and disaccharides) which are based on our analysis method including 24.9 g·L^−1^ glucose, 3.0 g·L^−1^ disaccharides (mainly sucrose), 4.3 g·L^−1^ xylose, and 0.4 g·L^−1^ arabinose (Table 2). Since the amount of lignin in banana peduncles was relatively low (6.16%), no pre-treatment processes were carried out and the hydrolysate was used directly for LA fermentation which reduces the cost of the process (Table 1). 

Lactic acid production using isolate A166 showed lag phase of ca 5 h and after 23 h most of the sugars were consumed and resulted in the production of 26.6 g LA·L^−1^, acetic acid 2.5 g·L^−1^ and the remaining unconsumed sugars were 3.1 g·L^−1^ (Figure 1). The calculated LA yield was 0.90 g LA·g^−1^ sugars, and the calculated productivity at the log phase was 3.61 g·L^−1^·h^−1^. Since there were no reports on LA production from banana peduncles, results are comparable to those obtained from two different types of banana organic waste, i.e., peel and flesh of un-matured banana by using strain *Lactobacillus bp Pentosus* using a simultaneous saccharification and fermentation (SSF) process [36]. The sugar content of 10% (*w*/*v*) was lower in the peels (4 g·L^−1^) and higher in the flesh (56 g·L^−1^), and the final concentrations of LA reached 4.8 g·L^−1^ and 50 g L^−1^, respectively. Our banana peduncle bagasse of 10% (*w*/*v*) produced more LA compared to peel and relatively lower than flesh banana (data not shown). 

### 2.3. Fermentation of Sugarcane Biomass

Hydrolysis of sugarcane of 2019 bagasse at 15% DM using a mixture of Accellerase BG and Cellic CTec2 enzymes and fermentation using SHCF process and isolate A166 for LA production were carried out. Since the 15% DM of 2018 bagasse showed the same trend as the 2019 harvest, the results of 2019 are the only presented data. Sugar content of sugarcane (15% DM) after enzymatic hydrolysis resulted in 54.5 g·L^−1^ total sugars (mono- and disaccharides) which are based on our analysis method (Table 2). As the lignin content of sugarcane of 2019 lignocellulosic bagasse was very low (2.79%), no pretreatments were carried out and the hydrolysate was used directly for LA fermentation. Lactic acid production using isolate A166 showed no lag phase and after 18h all the sugar was consumed and resulted in the production of 46.5 g LA·L^−1^, acetic acid 1.7 g·L^–1^ and the remaining unconsumed sugars were 0.3 g·L^−1^ (Figure 2). The calculated LA yield was 0.88 g LA·g^−1^ sugars, and the productivity at the log phase was 6.67 g LA·L^−1^·h^−1^. During the lag phase, the strain A166 breaks down inhibitors such as fufural or HMF. While low concentrations of it persist within the medium, the strain will not start growing. In our case the lag phase was negligible because of the adapted used A166 isolate. We have already been able to determine this with other lignocellulose substrates and have also measured that in previous work [42]. 

Van Der Pol et al., 2016 [43] applied acid pretreatment, steam explosion, and SSF to 20% *w*/*w* DW sugarcane bagasse using *B. coagulans* DSM2314 (consumed xylose and glucose simultaneously), and indicated that LA production from lignocellulosic hydrolysates was 64.1 g LA·L^−1^ (with 80% of yield and 0.78 g LA·L^−1^·h^−1^. of productivity). In another study, conducting simultaneous saccharification and co-fermentation (SSCF) with an initial pretreatment of the sugarcane bagasse using *L. pentosus* showed total consumption of both, xylose and glucose, producing 65.0 g LA·L^−1^, and 0.93 g·g^−1^ of yield, where the productivity reached 1.01 g LA·L^−1^·h^−1^ [3]. In another work, sugarcane bagasse hydrolysate obtained after pretreatment with dilute acid and alkaline was fermented using SHCF process for LA production by *Lactobacillus* spp, and resulted in a LA production and productivity of 42.5 g LA·L^−1^ and 1.02 g LA·L^−1^·h, respectively. Our results using the SHCF without any pretreatment showed similar results concerning the amount of LA and an even better yield and productivity, thus saving the cost of pretreatment and the production of fermentation inhibitors (Figure 2).

### 2.4. Fermentation of Carob

Preliminary studies were conducted to test the enzymatic hydrolysis using a mixture of Accellerase BG and Cellic CTec2 compared to unhydrolyzed of carob biomass at 20% and 30% DM. The fermentation potential was performed using two different *B. coagulant* isolates A107 and A559 for LA production which was carried out using SHCF process. The results showed that isolate A107 was fast growing with a short lag phase; therefore, we decided to use it for further fermentation studies (data not shown). Hydrolysis of 30% carob biomass resulted in the increase of total sugars by more than 17%, and the amount of LA increased by more than 25% (66.2 g LA·L^−1^ vs 51.6 g LA·L^−1^). Therefore, we decided to continue our experiments using 30% DM with enzymatic hydrolysis and testing the potential of isolate A107 to produce LA with and without the addition of yeast extracts. 

Sugar types and content of carob (30% DM) biomass after enzymatic hydrolysis used for LA production with or without yeast extract were similar, where supplementation with yeast extract resulted in total sugars of 74.3 g·L^−1^, including 42.0 g·L^−1^ glucose, 1.9 g·L^−1^ disaccharide, and 30.4 g·L^−1^ fructose. However, without yeast extract, total sugars were 68.9 g·L^−1^ including 38.9 g·L^−1^ glucose, 1.9 g·L^−1^ disaccharide, and 28.1 g·L^−1^ fructose. Previous studies show that the geographical origin and ripening stage of carob affects sugar yield quantitatively yet not qualitatively, where the main sugars identified in whole pods during development were sucrose, fructose and glucose [17]). Throughout the maturity stages the variation in sugar content was expressed by a slight reduction of fructose and glucose content and increasing the accumulation of sucrose in the carob biomass [17]. Our results showed that in our mature carob lignocellulosic biomass after syrup extraction following enzymatic hydrolysis the main sugars were glucose then fructose, and the total sugars with or without protein hydrolysis showed no significant differences (Table 3). 

Lactic acid production using isolate A107 with yeast extract showed short lag phase of 2 h and after 19 h most the sugars were consumed and resulted in the production of 54.8 g LA·L^−1^, 0.4 g·L^−1^ acetic acid and the remaining unconsumed sugars were 7.9 g·L^−1^ (Figure 3).

The calculated LA yield was 0.83 g LA·g^−1^ sugars, and the calculated productivity at the log phase was 3.2 g LA·L^−1^·h^−1^. Lactic acid production using isolate A107 without yeast extract showed a longer lag phase of 4 h and after 35 h most of the sugars were consumed and resulted in the production of 51.4 g LA·L^−1^, acetic acid 0.6 g·L^−1^ and the remaining unconsumed sugars were 6.4 g·L^−1^ (Figure 3). The calculated LA yield was 0.82 g LA·g^−1^ sugars, and the productivity at the log phase was 1.95 g LA·L^−1^·h^−1^. Previous studies using different bacterial isolates to produce LA from carob pods syrup (not including the lignocellulosic biomass) showed that supplementation of yeast extract with or without beef extract into the medium were very important to increase LA or bioethanol production and yields [29,44,45]. Our results indicate that we can save yeast extract supplementation into carob bagasse fermentation and still obtain relatively high LA productivity. It is an important to avoid yeast extract supplementation to the fermentation process not only reduce the cost as an advantage, but will also result in a product with less impurities requiring removal. However, we understand that the fermentation process without yeast extract supplementation is longer, and this should be taken into account when applying in industrial process. In addition, as in our case, the proteinases are more favorable for the downstream process.

The effect of sugar hydrolysis followed by protein hydrolysis compared to without hydrolysis was carried out in duplicates on LA production from 30% carob biomass which showed that the amount of total sugars ranged between 68.8 and 70.3 g·L^−1^, respectively, with no significant differences between the two (Table 3). The final amount of LA produced with protein hydrolysis compared to without hydrolysis treatment were similar 50.7 and 51.3 g LA·L^−1^, respectively and no significant differences were found (Figure 4). The calculated LA yields of protein hydrolysis compared to without hydrolysis were 0.87, and 0.82 g LA·g^−1^ sugars, and the productivity at the log phase were 3.49 and 2.98 g LA·L^−1^·h^−1^, respectively. In addition, the protein hydrolysis treatment showed faster growth rate and the fermentation process was completed within 26 h, compared to without protein hydrolysis the fermentation process was completed after longer period of 32 h (Figure 4). Therefore, the enzymatic sugar hydrolysis followed by protein hydrolysis was selected for further fermentation experiments using the 35 L pilot scale fermentation for carob biomass.

### 2.5. Larger Scale Fermentation for Carob

Sugar types and content of carob (30% DM) biomass after enzymatic hydrolysis used for LA production without yeast extract using the SHCF process of one liter fermenter compared to 35 L pilot scale were similar and resulted in total sugar of 67.9 g·L^−1^ and 70.6 g·L^−1^, respectively (Table 4).

Lactic acid production using isolate A107 of one liter scale fermenter showed almost no lag phase and after 26 h all the degradable sugars were consumed and resulted in the production of 54.8 g LA·L^−1^, acetic acid 0.4 g·L^−1^ and the remaining unconsumed sugars were 9.74 g·L^−1^ (Figure 5). The calculated LA yield was 0.89 g LA·g^−1^ sugars, and the productivity at the log phase was 3.28 g LA·L^−1^·h^−1^. Lactic acid production using the 35 L pilot scale fermenter showed almost no lag phase, however the growth rate was slower and only after 50 h of fermentation most of the degradable sugar was consumed and resulted in the production of 48.7 g LA·L^−1^, no acetic acid was produced and the remaining unconsumed sugars were 12.7 g·L^−1^ (Figure 5).

The calculated LA yield was 0.84 g LA·g^−1^ sugars, and the productivity at the log phase was 2.30 g LA·L^−1^·h^−1^. As our work is the first published data on the use of carob bagasse for LA production, we compared our results with those obtained using carob syrup. The fermentation of carob pod syrup was conducted by *L. plantarum* for production of LA at a concentration of 5% syrup using five fermentation conditions: 1) control conducted in carob syrup, 2) fermentation by addition of 10 g·L^−1^ of beef extract, 3) culture enriched by 10 g·L^−1^ of beef extract and 5 g·L^−1^ of yeast extract, 4) fermentation supplemented by these two components and 2 g·L^−1^ of K2HPO4 and 5) the fermentation treated by these three substances and 2 g·L^−1^ of triammonium citrate) [44]. The highest production of 49.34 g LA·L^−1^ was obtained when the culture was enriched by 10 g·L^−1^ of beef extract followed by syrup supplemented by all components 46.29 g LA·L^−1^ [44]. The highest production level of 49.34 g LA·L^−1^ was obtained using carob syrup supplemented with 10 g·L^−1^ of beef extract. In our current work, using the 35 L pilot scale with 30% (*w*/*v*) carob bagasse without yeast extract, we were able to produce 48.7 g LA·L^–^ using just a carob waste (Figure 5).

The long log phase in our batch pilot scale could be attributed to mass transfer, mixing and/or in-homogeneities inside the reactor. Recent studies suggested that continuous fermentation process is advantageous for high LA productivity due to dilution of the produced LA in the broth resulted from feeding new medium and high operational stability by achieving steady state of cell growth. The LA production, and sugar consumption, compared with batch or fed-batch fermentation processes where usually difficult to sustain homogeneities in large scales [15,16,46]. It was suggested that continuous SSF with enzyme addition and cell recycle can solve the problems caused by feedback, substrate and end-product inhibition compared to batch or fed-batch fermentation while resulting in higher LA productivities, yield, and concentration [47]. Several recent studies showed that continuous fermentation with cell recycle resulted in a higher LA productivity, and when the cells reached a steady-state, a high LA level was maintained for longer time and the utilization of the sugars was at maximum to produce LA (7,15). In addition, these studies have shown that, the lag phase in the cells was reduced, the operational cost was lower, and resulted in the increase of LA production (7,15). The pilot scale using carob biomass which is just a lignocellulose bagasse when fermented even without yeast extract supplementation resulted in high yield of 0.84 g LA·g^−1^ sugars, and productivity of 2.30 g LA·L^−1^·h^−1^ which is very promising process for industrial production of LA. Based on our results we suggest avoiding yeast extract supplementation into the fermentation process will save us dealing with impurities during downstream process, which is highly advantageous. In addition, based on our preliminary calculations, using proteinases is cheaper compared to yeast extract.

## 3. Materials and Methods

### 3.1. Feedstock and Biomass Analysis

Peduncles of *Musa cavendishi* bananas were collected from a Galilee plantation packing house in Beit Shean region, Israel. Fresh banana peduncles were randomly selected and manually separated from the cluster and dried under shade for 5 days at 30 °C. The peduncles were moved to a sugarcane press and a corn grinder for juice extraction which was used for ethanol production. The banana peduncle biomass remaining after juice extraction was dried in air for 2 days at 30 °C, then in a vacuum oven for 24 h at 50 °C and stored at room temperature of 25 °C until use.

Sugarcane (*S. officinarum*) was collected from a constructed wetland used to treat municipal wastewater from Sachnin, Galilee region, Israel. The juice was extracted using sugarcane press and a corn grinder. The remaining biomass was dried in air for 2 days at 30 °C, then in a vacuum oven for 24 h at 50 °C and stored at room temperature of 25 °C until use.

Carob (*C. siliqua* L.) pods were collected from Nazareth, the Galilee region, Israel, dried in the shade and used to extract the syrup. The dried pods were grinded using local grinder; the grinded pod biomass was infused in a cold water bath with gentile agitation for 24 h at room temperature of 25 °C to extract the sugars. The carob biomass was washed gently with water, filtered using a cloth filter and the remaining biomass was dried in air for 2 days at 30 °C and then in a vacuum oven for 24 h at 50 °C, grinded using a corn grinder and stored at 25 °C until use.

The main components of the different lignocellulosic bagasse were determined using the laboratory analytical protocol (LAP) developed by the National Renewable Energy Laboratory (NREL) using 100 g of each: carob, banana peduncles and sugarcane bagasse.

### 3.2. Lignocellulose Enzymatic Hydrolysis

Hydrolysis of the different lignocellulosic biomass were carried out at adjusted pH of 5 where a mixture of two commercial enzymes were used, the cellulase Cellic^®^ CTec2 (Novozyme) at dosage of 5.4% *v*/*w* (mL enzyme·g^−1^ biomass) and supplemented with Accellerase^®^ BG (DuPont) at 0.25 mL·g^−1^ biomass (both as recommended by the manufacturers) and based on previous work (The mixture was stirred (800 rpm) at 50 °C for 24 h using a 2 L BIOSTAT bioreactor (Sartorius AG, Göttingen, Germany). To determine total sugars (glucose, xylose, arabinose and sucrose), samples were taken aseptically.

In order to test the influence of protein hydrolysis on LA production, 30% carob biomass and water were added in 2 L sterile reactors and the sugar hydrolysis conditions were set as previously described by adding both enzymes. The enzymatic hydrolysis of the lignocellulose components was stopped after 24 h. Subsequently, the conditions for protein hydrolysis were set (the same as fermentation conditions). The pH value was adjusted to 6.0 and the temperature was increased to 52 °C. A mixture of 0.0067% (*v*/*w*) Neutrase 0.8L and 0.0067% (*v*/*w*) Flavourzyme 500L were added. After 30 min of incubation the pre-culture was added. To test the effect of protein hydrolysis on the fermentation process, another fermentation process without this hydrolysis step but with the same fermentation condition was inoculated at the same time. For this purpose, a common pre-culture medium was used.

### 3.3. Fermentation

The strains *Bacillus coagulans* isolate A107 and A166 which are available at the Leibniz Institute for Agricultural Engineering and Bioeconomy (Potsdam, Germany) were used for the fermentation processes. The inoculum preparation was carried out in 250 mL flasks containing 60 mL of De Man Rogosa Sharpe (MRS) broth (Merck, Germany) with Everzit Dol (Evers, Germany) dolomite as a buffer, and cultivated for 10–16 h before being used in the different fermentation experiments. Isolate A166 was isolated from hemp leaves and A107 was isolated from rapeseed extraction meal, where the cultivation medium was MRS broth (Merck, Germany) with 0.67 g Everzit Dol (Evers, Germany) dolomite as a buffer solution. The cultivation conditions of both strains were at 52 °C, initial pH of 6.0, growth for 16 h. The storage conditions were performed in CRYOINSTAND Yellow 50-Cryotubes 1 mL (VWR Chemicals) at −80 °C.

The lab scale fermentations for the 3 different biomasses were carried out at 52 °C, with stirring at 400 rpm. The banana peduncle biomass (of 10% or 15% (*w*/*v*) dry matter, DM), sugarcane bagasse from biomass collected from two harvesting years 2018, 2019 (15% (*w*/*v*) DM), or carob biomass (20 or 30% (*w*/*v*) DM) were studied using MRS broth, the same as the cultivation of the inoculum. Banana peduncle biomass above 15% (*w*/*v*) DM was too much viscous in the fermenter; therefore, we decided to use the same amount from sugarcane (15%) in order to make comparison between both. The pH was adjusted to 6.0 with 20% (*w*/*w*) NaOH, for all the studied strains. Inoculum volume used was 6% (*v*/*v*) which is a standard amount we have previously used for similar substrates lignocellulos which showed optimal growth rates [7,42]. For the batch fermentations we used the strain *B. coagulans* isolate A107 for carob, *B. coagulans* isolate A166 for banana and sugarcane bagasse where all the experiments were performed in triplicates using 2 L (one liter working volume) of BIOSTAT bioreactors (Sartorius AG, Göttingen, Germany) at 52 °C, using different biomass supplemented with yeast extract of 10 g per liter. Stirring was set to 400 rpm and the pH was automatically adjusted to 6.0 using 20% (*w*/*w*) NaOH. Samples were withdrawn aseptically for the analysis of the different sugars (glucose, xylose, arabinose and sucrose) and LA concentrations. Samples were inactivated using water bath at 95 °C for 20 min. After inactivation, the collected samples were stored at –20 °C until being analyzed. 

### 3.4. Analytical Methods

The determination of different sugars and LA production from each fermentation sample was carried out using HPLC (Dionex, Sunnyvale, CA, USA), equipped with a Eurokat H column (300 mm × 8 mm × 10 µm, Knauer, Berlin, Germany). An aqueous solution of 5 mM H_2_SO_4_ was used as the mobile phase, at a flow rate of 0.8 mL min^−1^. Injection volume was 10 µL and the detection was carried out by a refractive index detector (RI-71, Shodex, Tokyo, Japan). The optical purity of the produced LA was also determined using the same HPLC (Dionex, USA), and a Phenomenex Chirex 3126 (150 9 4.6 mm ID, Phenomenex, Torrance, CA, USA) column, temperature of 30 °C, with 1 mM Cu_2_SO_4_ as eluent at a flow rate of 1 mL min^−1^. The detection of the different components was carried out using an UV detector.

### 3.5. Larger Scale Fermentation for Carob

Due to the limitation in carob biomass, the scale up was carried out running only two bench scale fermentations (50 L BIOSTAT UD bioreactors, B-Braun Biotech, Melsungen, Germany), where each containing 35 L of carob biomass at 30% (*w*/*v*) DM to produce LA using the strain *B. coagulans* isolate A107. In addition, a 2 L (one liter working volume) of BIOSTAT bioreactors (Sartorius AG, Göttingen, Germany) was used for comparison purposes. The preculture for inoculation process was carried out in 2 steps. In the first step we used a shaking flask with 40 mL MRS and Everzit Dol as a buffer without pH control, and in the second stage with 2 L synthetic medium of glucose, fructose and sucrose and yeast extract. The cultivation process was carried out in the 35 L fermenters with pH control. Carob biomass hydrolysis was carried out for 24 h as previously described, and then protein hydrolysis was carried out using 0.0067% (*v*/*w*) Neutrase 0.8 L and 0,0067% (*v*/*w*) Flavourzyme 500 L and incubation at 52 °C for 30 min. The protein hydrolysis was carried out based on previous experiments which showed that this process enhances growth rate of the *B. coagulans* (see results). The fermentation process for the 2 pilot scales and the 2 L lab scale were carried out using the strain *B. coagulans* isolate A107. Sampling and analysis were carried out as previously described.

### 3.6. Statistical Analysis

The averages, standard deviation and statistical analysis were performed using PRISM 8 statistical software. The one-way ANOVA statistical analysis was used to test significance in the study.

## 4. Conclusions

The LA production from bagasse of banana peduncles, sugarcane or carob showed very promising results where carob and sugarcane were the best biomass. We show that a high % of DM can be used during the fermentation process. However, despite high LA productivity, using banana allowed for a maximum solid of 15% due to high viscosity. Using carob biomass for LA production using SHCF process showed that no yeast extract addition is required. The pilot scale of 35 L using carob biomass without yeast extract resulted in high yield of 0.84 g LA·g^−1^ sugars, and productivity 2.30 g LA·L^−1^·h^−1^.

## Figures and Tables

**Figure 1 molecules-25-02956-f001:**
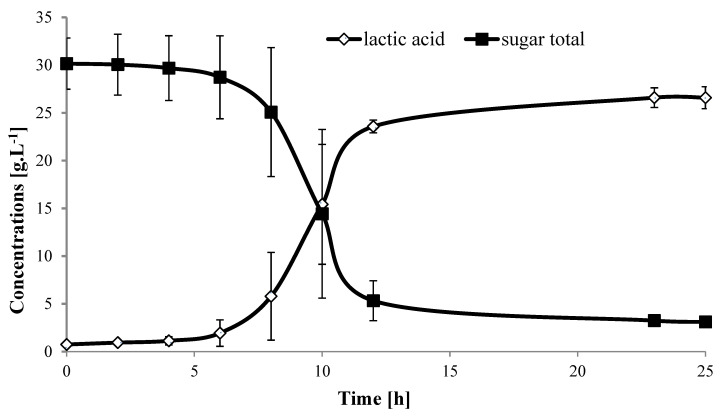
Lactic acid production and remaining sugars trend during 25 h fermentation after hydrolysis of banana peduncles biomass (15% DM) using isolate A166. Each data point represents the mean of 3 replicates ± SD.

**Figure 2 molecules-25-02956-f002:**
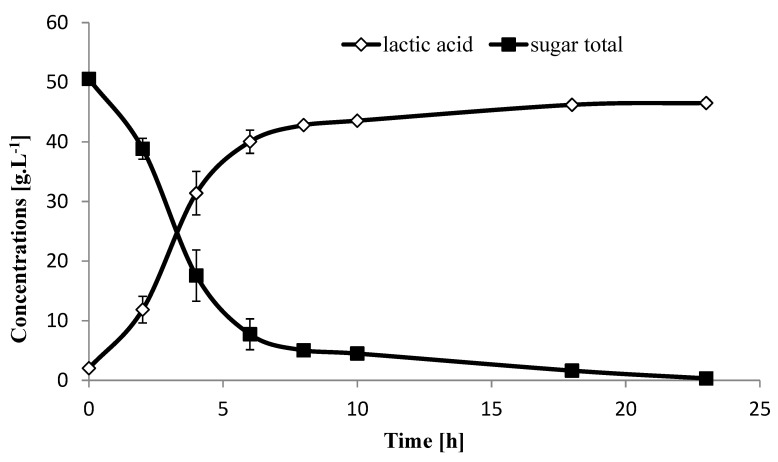
Lactic acid production and remaining sugars trend during 24 h fermentation after hydrolysis of sugarcane biomass (15% DM) of 2019 harvest using isolate A166. Each data point represents the mean of 3 replicates ± SD.

**Figure 3 molecules-25-02956-f003:**
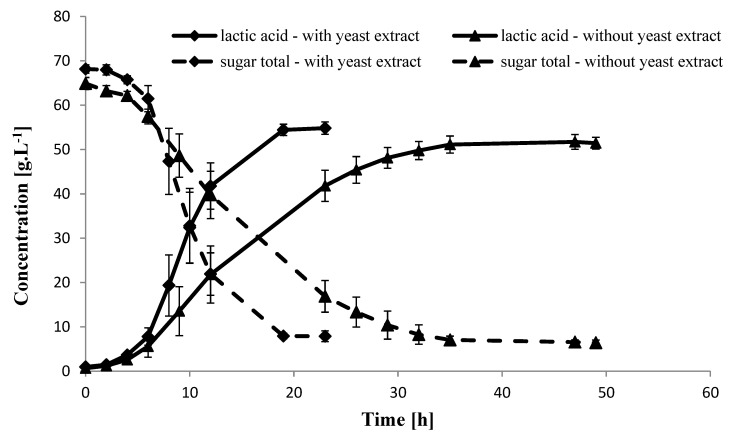
Lactic acid production and remaining sugars trend during fermentation after hydrolysis of carob biomass (30% DM) with or without yeast extract using isolate A107. Each data point represents the mean of 3 replicates ± SD.

**Figure 4 molecules-25-02956-f004:**
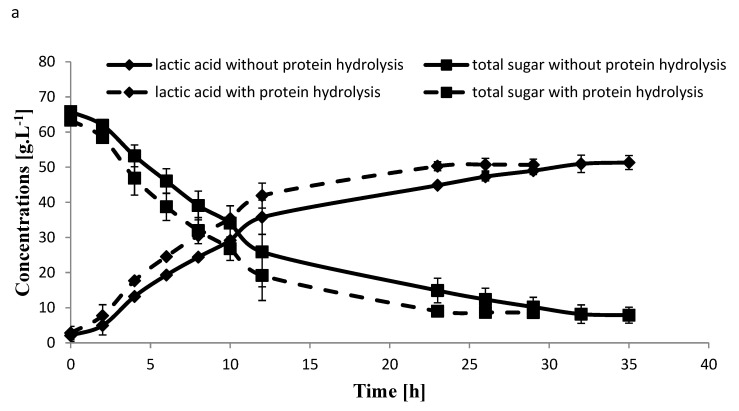
Lactic acid production and remaining sugars trend during fermentation after enzymatic sugar hydrolysis of carob biomass (30% DM) followed with or without protein hydrolysis conducted in one liter fermenter without yeast extract using isolate A107 (**a**) and comparison end of total lactic acid, acetic acid, and total sugars after fermentation (**b**). Each data point represents average of two replicates.

**Figure 5 molecules-25-02956-f005:**
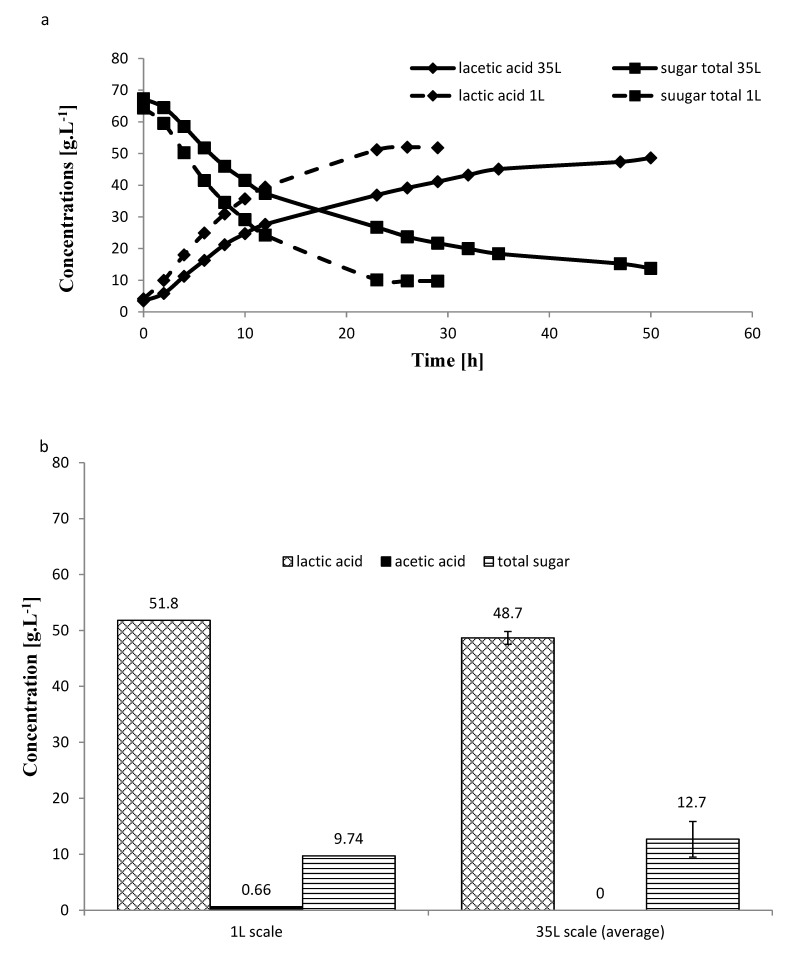
Lactic acid production and remaining sugars trend during fermentation after sugar hydrolysis followed by protein hydrolysis of carob biomass (30% DM) conducted in one liter fermenter compared to 35 L pilot scale without yeast extract using isolate A107 (**a**) and comparison end of total lactic acid, acetic acid, and total sugars after fermentation without yeast extract (**b**). Each data point represents the mean of 2 replicates ± SD of the pilot scale and one replicate for the 1L scale.

**Table 1 molecules-25-02956-t001:** Composition of carob, banana peduncle and sugarcane biomass (% of dry matter, DM).

	DM 105 °C [%]	Proteins [%DM]	Sugars [%DM]	Cellulose [%DM]	Hemicellulose [%DM]	Lignin [%DM]
**Banana**	93.2	8.2	0.1	35.8	20.7	6.16
**Sugarcane**	91.1	12.5	11.3	27.9	25.6	2.79
**Carob**	85.8	9.3	27.7	19.0	0.35	28.4

**Table 2 molecules-25-02956-t002:** Sugar types, content and total sugars (g·L^−1^) of banana peduncles (15% DM) and sugarcane biomass (15% DM) after enzymatic hydrolysis. Each data point represents the mean of 3 replicates ± SD.

	Glucose (g·L^−1^)	Disaccharides (g·L^−1^)	Xylose (g·L^−1^)	Arabinose (g·L^−1^)	Total Sugars (g·L^−1^)
**Banana**	24.9 ± 1.6	3.0 ± 0.4	4.3 ± 0.0	0.4 ± 0.0	32.6 + 2.1
**Sugarcane**	30.6 ± 0.7	3.3 ± 0.0	19.4 ± 0.4	1.3 ± 0.1	54.6 + 1.2

**Table 3 molecules-25-02956-t003:** Sugar types, content and total sugars (g·L^−1^) of carob biomass (30% DM) used for LA production after enzymatic hydrolysis followed by protein hydrolysis compared to without hydrolysis process. Data are average 2 ± SD replicates. Different letters in each column indicate significant difference at *p* < 0.05.

	Glucose (g·L^−1^)	Disaccharides (g·L^−1^)	Fructose (g·L^−1^)	Total Sugars (g·L^−1^)
**With Protein hydrolysis**	36.6 ± 0.3a	3.0 ± 1.1a	27.2 ± 0.2a	66.8 + 1.6a
**Without Protein hydrolysis**	38.8 ± 0.1b	3.0 ± 1.5a	28.5 ± 0.0b	70.3 + 1.6a

**Table 4 molecules-25-02956-t004:** Sugar types, content and total sugars (g·L^−1^) of carob biomass (30% DM) used for LA production after enzymatic hydrolysis using one liter fermenter or 35 L pilot scale.

	Glucose (g·L^−1^)	Disaccharides (g·L^−1^)	Fructose (g·L^−1^)	Total Sugars (g·L^−1^)
**One liter fermenter**	36.8	3.8	27.3	67.9
**35 L pilot scale**	38.8	3.8	28.0	70.6

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
