# Peer review of "Production of Lactic Acid from Carob, Banana and Sugarcane Lignocellulose Biomass"

_molecules, 2020, doi:10.3390/molecules25132956_

Round 1

Reviewer 1 Report

The proposed title it is recommended to be revised because this in not according with the aims and obtained results. The obtaining of the polylactic acid as a biodegradable plastic was not studied in this paper.

The innovation of the study and principal approaches must be very clearly highlighted in the sections aims of the study and conclusions.

In the section Material and methods some aspects regarding fermentation must be improved as follow:

  • Details about the strains of Bacillus coagulans used in fermentation, their isolation sources, cultivation and the preservation of stock cultures.
  • Details about inoculum concentration evaluation and what was the reasons to be used of 6% concentration. What was the reason for use of 6% inoculum?
  • What was the reason for use MRS medium for cultivation of the inoculum?
  • In subsection 2.5. Larger scale fermentation for carob, the inoculation process is not described at all.
  • The information about the enzyme’s preparation used for substrate hydrolysis must be completed, i.e. the enzyme units and their definition. Also, their sources it is important to be specified.
  • A statistical analysis section must be inserted and use for results interpretation.

The comparison of the results with the scientific literature data is very few used in the manuscript. This aspect must be strong improved and carefully revised.

Author Response

The proposed title it is recommended to be revised because this in not according with the aims and obtained results. The obtaining of the polylactic acid as a biodegradable plastic was not studied in this paper. The title was corrected accordingly

The innovation of the study and principal approaches must be very clearly highlighted in the sections aims of the study and conclusions.

In the section Material and methods some aspects regarding fermentation must be improved as follow:

  • What was the reason for use MRS medium for cultivation of the inoculum? MRS was the isolation medium. We used MRS from previous works that our strains grow well in this medium. There was no further optimization. Details were added in the manuscript.
  • In subsection 2.5. Larger scale fermentation for carob, the inoculation process is not described at all. The inoculation details were added in section 2.5 as requested.
  • The information about the enzyme’s preparation used for substrate hydrolysis must be completed, i.e. the enzyme units and their definition. Also, their sources it is important to be specified. The information on the enzymes were added in section 2 Lignocellulose Enzymatic Hydrolysis

  • A statistical analysis section must be inserted and use for results interpretation. Statistical analysis section was added

The comparison of the results with the scientific literature data is very few used in the manuscript. This aspect must be strong improved and carefully revised.  This is very important point and it was improved and corrected in the section of Results and Discussion.

Reviewer 2 Report

The authors investigated the production of LA from carob, banana and sugarcane lignocellulose biomass. And they explored the conditions for industrial production of LA, providing a more economical promising process with comparable productivity. Although practical, the language and details in the manuscript need to be further optimized.

  1. Page 1, line 24, the author states that “Carob showed that addition of yeast extract 24 compared to without yeast extract caused higher productivity of 3.2 and 1.95 g.L-1.h-1, respectively ” What exactly does the productivity here refer to (the yield of total, LA or sugar) ? Perhaps the higer productivity of yeast compared than yeast-free should be the difference between the two? What does the “respectively ” here refer to? It's so confusing.

  1. The logic of the introduction is incoherent, the narration of the three biomass is prolix, and the focus and significance are not prominent.

  1. The parameters of some experiments should be given in more detail. Page3, line 115, what is the room temperature exactly? There is a great temperature difference throughout the year. Page3, line 118, the specific conditions for drying in the air include the number of days of drying? Page3, line 134, “30% carob biomass” 30% of what? Page4, line 177, What is the temperature of incubation?

  1. The author suggests that “Using carob biomass for LA production using SHCF process showed that there is no need to add yeast extract. ” Although a high productivity can be maintained without adding yeast, it takes longer to obtain a yield that matches the yeast addition strategy. How is the time cost evaluated?

  1. Page 5, line 204, The reasons for choosing 15% DM under the same trend should be explained.

6.The influence of the lag phase by using isolate A166 in lactic acid production should be discussed and analyzed.

  1. Some sentences are too long to interpret, and (likely) grammatically incorrect, please break it up, also, fix the grammar. For example: Page 1, line 38-42; Page 2, line 55-59; Page 7, line 249-252; Page 12, line 354-357 , etc.

  1. There are some format errors in the textand references. For example:Page 5, Table 2; Page13, line366, ref.1; Page14, line456, ref.32 (the titles of these references are capitalized, inconsistent with the rest); Page14, line444, ref.27 (punctuationmissing).

Author Response

The authors investigated the production of LA from carob, banana and sugarcane lignocellulose biomass. And they explored the conditions for industrial production of LA, providing a more economical promising process with comparable productivity. Although practical, the language and details in the manuscript need to be further optimized.

- Page 1, line 24, the author states that “Carob showed that addition of yeast extract 24 compared to without yeast extract caused higher productivity of 3.2 and 1.95 g.L-1.h-1, respectively ” What exactly does the productivity here refer to (the yield of total, LA or sugar) ? Perhaps the higer productivity of yeast compared than yeast-free should be the difference between the two? What does the “respectively ” here refer to? It's so confusing.

The whole sentence was corrected where the productivity of LA was changed to gLA.L-1.h-1 in the whole text

- The logic of the introduction is incoherent, the narration of the three biomass is prolix, and the focus and significance are not prominent.  

  • The introduction was changed to meet the coherent and show the significance why the 3 biomass were selected

- The parameters of some experiments should be given in more detail. Page3, line 115, what is the room temperature exactly? There is a great temperature difference throughout the year. Page3, line 118, the specific conditions for drying in the air include the number of days of drying? Page3, line 134, “30% carob biomass” 30% of what? Page4, line 177, What is the temperature of incubation?

  • All parameters were corrected throughout he manuscript

- The author suggests that “Using carob biomass for LA production using SHCF process showed that there is no need to add yeast extract. ” Although a high productivity can be maintained without adding yeast, it takes longer to obtain a yield that matches the yeast addition strategy. How is the time cost evaluated?

  Response: The time is important but with Carob biomass which is rich with sugars we save yeast extract supplementation which is expensive for mass production, and our results showed that we have obtained high yield and productivity without supplementation of yeast extract not far away from those experiments where yeast extract was added.  In addition without yeast extract the downstream process with less impurity. This argument was added into the manuscript.

- Page 5, line 204, The reasons for choosing 15% DM under the same trend should be explained.

We have explained in the text concerning the viscosity which caused by the banana peduncles when the % of biomass was high such as at 20%. And also in the conclusions.

- The influence of the lag phase by using isolate A166 in lactic acid production should be discussed and analyzed. This issue was discussed in section 3.3 Fermentation of sugarcane biomass

- Some sentences are too long to interpret, and (likely) grammatically incorrect, please break it up, also, fix the grammar. For example: Page 1, line 38-42; Page 2, line 55-59; Page 7, line 249-252; Page 12, line 354-357 , etc. The sentences were shortened and the grammar was fixed, and the whole manuscript was edited.

- There are some format errors in the textand references. For example:Page 5, Table 2; Page13, line366, ref.1; Page14, line456, ref.32 (the titles of these references are capitalized, inconsistent with the rest); Page14, line444, ref.27 (punctuationmissing).  The capitalized and the punctuation were corrected

Reviewer 3 Report

The authors of the present study examine the fermentative production of lactic acid using hydrolysates of banana peduncles, and sugarcane and carob bagasse. Their findings can contribute to a more efficient exploitation of food and plant biomass for the production of chemicals of added value, in the frame of a cyclic economy. However, before the manuscript is accepted the authors should address the following comments:

  1. It is not clear why the authors supplemented Cellic CTec2 by a beta-glucosidase for the hydrolysis of the 3 substrates, since the former already contains a high level of beta-glucosidases.
  2. Page 6, line207: explain “disaccharide”
  3. Table 2: total sugars is presented as the sum of the previous columns. Usually total sugars are determined independently, as there may be additional components such as trisaccharides, mannoses and galactoses, not taken into account by this method.
  4. Page 6, lines 228-230: in general, pretreatment to remove lignin is done prior the enzymatic hydrolysis, so this phrase should be corrected.
  5. Page 7, lines 258-262: since supplementation with yeast extract was done at the fermentation stage, it is not clear what the authors mean in this phrase, where they imply addition of yeast extract upon the hydrolysis step.
  6. Figure 5, add sd for the 35L fermentation in the graph. The comparison would be more reliable if there were duplicates for the 1lt fermentation as well.
  7. The authors show that the addition of YE or protein hydrolysis before fermentation both result in higher productivity and shorter fermentation times for similar lactic acid yield. They thus suggest that there is no need to add YE, and perform a step of protein hydrolysis using commercial proteases. They should however comment on whether the use of enzymes instead of YE is a cheaper approach, thus worth applying it in large scale fermentations.

Author Response

The authors of the present study examine the fermentative production of lactic acid using hydrolysates of banana peduncles, and sugarcane and carob bagasse. Their findings can contribute to a more efficient exploitation of food and plant biomass for the production of chemicals of added value, in the frame of a cyclic economy. However, before the manuscript is accepted the authors should address the following comments:

  1. It is not clear why the authors supplemented Cellic CTec2 by a beta-glucosidase for the hydrolysis of the 3 substrates, since the former already contains a high level of beta-glucosidases.  This is correct, but since our optimization process showed better hydrolysis data using a mixture we decided to use both. This was explained in the Material and method section.
  2. Page 6, line207: explain “disaccharide”. Disaccharide were defined
  3. Table 2: total sugars is presented as the sum of the previous columns. Usually total sugars are determined independently, as there may be additional components such as trisaccharides, mannoses and galactoses, not taken into account by this method.This is correct but based on our method this include all the sugars we were able to detect in our biomass.  A sentence was added in the text related to total sugars obtained using our analysis method.
  4. Page 6, lines 228-230: in general, pretreatment to remove lignin is done prior the enzymatic hydrolysis, so this phrase should be corrected.  The sentence was corrected.
  5. Page 7, lines 258-262: since supplementation with yeast extract was done at the fermentation stage, it is not clear what the authors mean in this phrase, where they imply addition of yeast extract upon the hydrolysis step.  This was corrected
  6. Figure 5, add sd for the 35L fermentation in the graph. The comparison would be more reliable if there were duplicates for the 1lt fermentation as well.  In Figure 5 we have already added the SD.
  7. The authors show that the addition of YE or protein hydrolysis before fermentation both result in higher productivity and shorter fermentation times for similar lactic acid yield. They thus suggest that there is no need to add YE, and perform a step of protein hydrolysis using commercial proteases. They should however comment on whether the use of enzymes instead of YE is a cheaper approach, thus worth applying it in large scale fermentations. This is an important point comments were added at the end of section 3.4 Fermentation of Carob & at the end of section 3.5 Larger scale fermentation for carob

Round 2

Reviewer 1 Report

Accept in present form.

Reviewer 2 Report

The authors have addressed the comments.